# How do humans and machine learning models track multiple objects through occlusion?

**Benjamin Peters**
Zuckerman Mind Brain Behavior Institute
Columbia University
New York, NY 10027
benjamin.peters@columbia.edu

**Eivinas Butkus**
Department of Psychology
Columbia University
New York, NY 10027
eivinas.butkus@columbia.edu

**Nikolaus Kriegeskorte**
Zuckerman Mind Brain Behavior Institute
and Departments of Psychology, Neuroscience, and Electrical Engineering
Columbia University
New York, NY 10027
n.kriegeskorte@columbia.edu

## Abstract

Interacting with a complex environment often requires us to track multiple task-relevant objects, not all of which are continually visible. The cognitive literature has focused on tracking a subset of visible identical abstract objects (e.g., circles), isolating the tracking component from its context in real-world experience. In the real world, object tracking is harder in that objects may not be continually visible and easier in that objects differ in appearance and so their recognition can rely on both remembered position and current appearance. Here we introduce a generalized task that combines tracking and recognition of valued objects that move in complex trajectories and frequently disappear behind occluders. Humans and models (from the computer vision literature on object tracking) performed tasks varying widely in terms of the number of objects to be tracked, the number of distractors, the presence of an occluder, and the appearance similarity between targets and distractors. We replicated results from the human literature, including a deterioration of tracking performance with the number and similarity of targets and distractors. In addition, we find that increasing levels of occlusion reduce performance. All models tested here behaved in qualitatively different ways from human observers, showing superhuman performance for large numbers of targets, and subhuman performance under conditions of occlusion. Our framework will enable future studies to connect the human behavioral and engineering literatures, so as to test image-computable multiple-object-tracking models as models of human performance and to investigate how tracking and recognition interact under natural conditions of dynamic motion and occlusion.

## 1   Introduction

Human vision parses the visual scene into objects. Task-relevant objects are tracked and represented in a persistent fashion that enables us to mind them even when they are out of our field of view or hidden behind an occluding object. This ability liberates our visual cognition from its tethers in

4th Workshop on Shared Visual Representations in Human and Machine Visual Intelligence (SVRHM) at the Neural Information Processing Systems (NeurIPS) conference 2022. New Orleans.

the sensory input and enables us to see the world in terms of its physical constituent components, providing a basis for prediction and causal insight (Peters & Kriegeskorte, 2021; Greff et al., 2020).

An influential class of behavioral tasks that has enabled cognitive scientists to probe this ability is multiple object tracking (MOT) (Pylyshyn & Storm, 1988; Scholl & Pylyshyn, 1999). MOT tasks are typically highly abstracted. Subjects are presented with a set of identical abstract objects (e.g., circles) in motion and cued to track a subset of them. At the end of each trial, the scene freezes and the subject indicates the cued objects. MOT is challenging for humans and computational models (Linsley et al., 2021; Vul et al., 2009). Humans can track up to three or four objects successfully even through brief occlusions (Intriligator & Cavanagh, 2001; Pylyshyn & Storm, 1988; Scholl & Pylyshyn, 1999; Yantis, 1992). Using abstract identical objects has enabled cognitive scientists to study the function of tracking in isolation from the broader process of real-world dynamic scene perception, where objects are diverse and tracking interacts with visual recognition to maintain persistent representations of the task-relevant objects. In the real world, object tracking is harder in that objects may be out of sight or hidden for longer periods. Real-world object tracking might also be easier in that diverse objects tend to be visually distinct enough to track them by their appearance. The visual system might therefore employ a mixture of strategies, tracking objects not only by maintaining and updating (even for currently occluded objects) their spatial positions and velocities but also by recognizing them by their appearance (Papenmeier et al., 2014; Zhou et al., 2010; Li et al., 2019).

Computer vision, with its focus on engineering applications, has engaged the MOT challenge in real-world settings (Milan et al., 2016; Geiger et al., 2012; Müller et al., 2018), building systems that can track objects in videos (e.g. pedestrians or cars in traffic). While tracking objects with robustness to periods of occlusion poses the same computational challenge to humans and machines, it is unclear to what extent current models employ computational strategies similar to those used by humans. Recent ML challenges involve a large number of pedestrians (Dendorfer et al., 2020) (up to $\sim 250$ pedestrians at the same time) widely exceeding the tracking capabilities of humans. On the other hand, supervised training on real-world images might bias the tracker towards less general solutions that might not reach human performance when required to generalize to unexpected challenges. In real-world scenarios, long stretches of occlusion of an object are rare compared to periods of at least partial visibility. Most tracked objects are detected with high confidence throughout their life-cycle (i.e., low noise regime), and objects are visually distinct enough to distinguish them by their appearance. Supervised ML object trackers might therefore be biased towards strategies that favor recognition in the current frame (spatial integration of evidence) over maintaining internal belief states about positions and velocities (temporal integration of evidence, as in a Kalman or particle filter). We might therefore expect these models to fail when similarity of appearance and/or longer stretches of occlusion render the recognition-based strategy underconstrained.

We here compare object tracking performance of four state-of-the-art supervised multiple object trackers with the performance of humans on the same task. We introduce a task combining object recognition demands and tracking designed to take steps toward bridging the gap between the real-world complexity of machine learning tasks and the abstraction of cognitive tasks. We replicated results from the human literature: a deterioration of tracking performance with the number and similarity of targets and distractors as well as the relatively unimpaired tracking performance through brief occlusion. We find that models displayed qualitatively different behavior from humans: their performance was independent of the number of objects, but deteriorated below human performance in the context of occlusions.

## 2 Methods

### 2.1 Models

We evaluate four state-of-the-art multiple-object tracking algorithms. Simple online and real-time tracking (**SORT**, Bewley et al. (2016)) uses a simple strategy for frame-by-frame tracking of multiple objects. Bounding boxes of object detections in each frame, provided e.g., by a Faster Region CNN (Ren et al., 2015), are the input to the tracker. On the first frame, the tracker initializes a series of tracklets, one for each detection, representing the objects to be tracked. Each tracklet's state (i.e., its bounding box center, height, aspect ratio, and their velocities) is tracked by a Kalman filter. In a new frame, the matches between all predicted bounding-box positions and all new detections are computed as intersection-over-union (IoU) to form an assignment cost matrix. The detections of

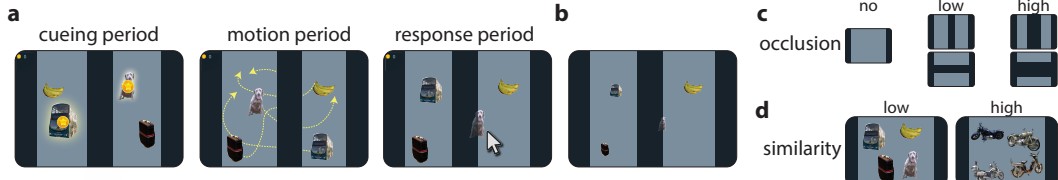

Figure 1: (a) Sequence of events (objects enlarged for visualization). (b) Object sizes at the scale of Experiment 1. Experimental factors occlusion level (c) and similarity (d).

the new frame are then associated with existing tracklets by minimizing the overall assignment cost using the Hungarian algorithm. Albeit a fairly simple algorithm, SORT achieved high performance on the MOT challenge (Milan et al., 2016; Lin et al., 2015) while enabling real-time application on standard hardware. SORT performs well when state uncertainty is low. However, during occlusions, SORT displays a high number of identity switches (i.e, multiple tracklets in turn track the same object over time). This is partly due to the linear dynamics model assumed by the Kalman filter, whose predictions are adequate for brief occlusions, but which become inaccurate for long occlusions given non-linear motion. **DeepSORT** (Wojke et al., 2017) is an extension of the SORT algorithm that also takes appearance information (an appearance descriptor from a CNN) into account during the association stage, thereby compensating for the high state uncertainty of the Kalman filter during occlusions. DeepSORT substantially reduced the number of tracklet switches. Tracking-by-detection approaches typically only use high-confidence detections of the object detector, leading to false negative detections for partially occluded objects, which in turn can lead to track termination and subsequent tracklet switches once the object reappears. Instead of discarding low-confidence detections, **ByteTrack** (Zhang et al., 2022) tracks almost every detection and uses similarity scores to distinguish between false positives and true detections. This strategy reduces the number of id switches compared to SORT and DeepSORT. **OC-SORT** (Cao et al., 2022) introduced several improvements over SORT/DeepSORT to better handle longer stretches of occlusion. For instance, one of the drawbacks of SORT is that the Kalman filter, which tracks the position and velocity of the bounding box, accumulates errors during stretches of occlusion. When the object reappears from occlusion it gets re-associated with its (memorized) track using appearance-based models. The Kalman filter belief about the objects position and velocity then needs to be reconciled with the newly detected position of the object. OC-SORT addresses this problem by implementing several strategies that give detections a much larger weight compared to the Kalman state in the association stage.

## 2.2 Task

The details of the object-tracking task were the same for humans and models unless stated otherwise (Figure 1a). At the beginning of each trial $N$ objects were presented. Phase 1 is the *cueing period* (Experiment 1: 1s, Experiment 2: 2.5s) in which the objects are stationary. For human subjects the target objects are highlighted in this period. Phase 2 is the *motion period* (five seconds here) in which the objects move according to random trajectories. We deliberately used a complex non-linear motion model rather than a linear motion model. In particular, angular motion direction perturbation were sampled at random time-points and the new direction was smoothly interpolated across multiple time-frames (intersection over union of an object's bounding boxes in adjacent time frames: 0.691, s.d.= 0.024 for Experiment 1 and 0.594, s.d.= 0.032 for Experiment 2). The objects then freeze for phase 3, the *response period*, in which the target objects are reported. Human subjects reported the targets by clicking on the objects. In occlusion trials, a large central opaque occluder covering either 20% or 40% of the field was superimposed to the scene. Stimuli were isolated objects extracted from the segmentation masks provided by the MS COCO challenge (Lin et al., 2015). For each of the 80 object categories, we randomly extracted 50 different exemplars, excluding fragmented objects. Object starting positions were sampled such that (1) all objects were unoccluded at the start of the trial and (2) all objects ended up in positions such that no two were closer to each other than a threshold distance. Objects which stopped their movement behind the occluder were revealed by turning the occluder semi-transparent (for humans) or moving their depth plane in front of the occluder (for models, which were not assumed to be familiar with semi-transparency) .

Our goal was to compare how well humans and models track multiple objects through occlusion using the appearance and spatiotemporal trajectory of objects. Human short-term memory for dynamic bindings, e.g., the binding between a visual object and a label or id, is highly limited. Rather than assigning labels or ids to objects in the cueing period and asking participants to reproduce those labels/ids during the response period, we decided to denote a subset of $T < N$ objects as "targets" that need to be tracked throughout the motion period (Pylyshyn & Storm, 1988; Merkel et al., 2019). To obtain tracking performance, we employed different strategies for humans and models.

## 2.3 Experiments

In Experiment 1, we presented 108 sequences with the following experimental factors: We varied the number of objects (4, 6, or 8). Half of the objects were targets in each case. A third of the trials contained no occluder, the remaining trials contained a rectangular occluder that extended over the full vertical (horizontal) length of the screen and either covered 20% or 40% of the horizontal (vertical) length (Figure 1c). We varied categorical similarity as a proxy for object similarity. Namely, objects within a trial could either all come only from one of the 80 categories (high similarity) or from different categories (low similarity) (Figure 1c). The factors, namely number of objects, occlusion level, occluder orientation (horizontal or vertical), and similarity were counter-balanced across trials. The maximum bounding box side length of an object (height or width) was 10% of the screen (Figure 1b).

Experiment 2 was identical to Experiment 1, with the following exceptions: First, there were four set size conditions: 6 objects - 2 targets, 8 objects - 4 targets, 12 objects - 2 targets, 14 objects - 4 targets. We also reduced the size of the objects to a maximum bounding box side length of 7%. Because of the higher visual load, we extended the cueing period to 2.5s to provide human observers with a sufficient amount of time to identify the cued objects.

## 2.4 Human behavior

For humans ($N = 10$ for each experiment), target objects were visually cued during the cueing period via a glowing outline and by overlaying an icon of a coin on top of each target object. During the response period, human participants clicked on $T$ objects, which they believed to be the initially cued target objects, after which the identity (i.e., target or non-target) of the selected objects was revealed by an animation that displayed and "collected" the coins of the target objects. There was no time-limit for human responses.

## 2.5 Model behavior

A major factor in object tracking performance is the quality of the object detector. We here focus on the tracking task rather than the ability to detect objects. Therefore, instead of relying on each model's detector, we provide models with ground truth bounding boxes for unoccluded objects in the current frame. This is akin to the public detections variant of the MOT challenge (Milan et al., 2016). Note that all models still used image input for appearance-based associations. We also provide the bounding box of the occluder as a detection, indicating to models that missing detections might be due to object-by-object occlusion rather than signal dropout.

To obtain responses from models, we analyzed whether the model consistently tracked target objects. First, we associated the ground-truth object positions with the model's tracklet positions (i.e., the model's belief about object positions) using the Hungarian algorithm. Second, we identified the target tracks as those tracks that the model assigned to the cued (target) objects on the first frame. A target object was considered to be successfully tracked if its final detection was assigned to one of the target tracks of the first frame. The model then "responded" to those final detections, that were assigned to a target track. In case less than $T$ detections were assigned to target tracks, the model randomly selected from the remaining detections (i.e., guessing) until $T$ responses were made. We report the average accuracy over 10 independent model evaluation runs.

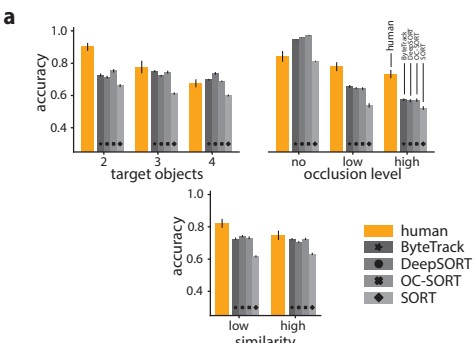
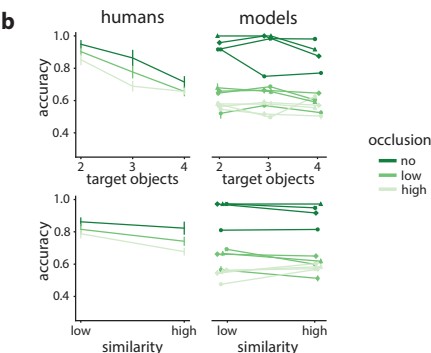

Figure 2: Results of Experiment 1. Response accuracy main effects (a) and selected interactions (b) for the factors of number of tracked objects, occlusion level, and object similarity. Error bar indicates standard error of the mean across participants (humans) or across repeated evaluation runs (models). See supplement for more interactions and model-specific plots (Figure 5).

# 3   Results[1]

In Experiment 1, we assessed the impact of set-size, occlusion, and object similarity on human and model performance. First, we replicated the *set-size* effect in humans. Namely, tracking more targets led to lower response accuracy for humans (Figure 2a, main effect set-size $F(2, 18) = 52.6$, $p < .001$). Models, in contrast, showed a non-significant set-size effect ($F(2, 6) = 3.5$, $p = .096$)[2]. Introducing *occlusion*, which renders parts of the object movements invisible to observers, led to lower performance in both humans and models (Figure 2a, humans: $F(2, 18) = 24.8$, $p < .001$, models: $F(2, 6) = 142.9$, $p < .001$). Humans and models however differed notably in their performance pattern: Models showed super-human performance during trials without occlusion, even at a small set size of four (i.e., two target objects) for which humans performed best. Strikingly, humans were better able to track objects through occlusion (low and high levels of occlusion) than any of the SOTA object trackers.

How do humans track multiple objects, in particular for complex non-linear object motion? A first indication can be gleaned from the effect of *object similarity*. In particular, increased similarity between objects reduced tracking performance for humans ($F(1, 9) = 31.3$, $p < .001$). This suggests that humans at least partially used object appearance information for tracking. Models do use appearance information for tracking per construction. However, the similarity manipulation we chose did not affect their performance ($F(1, 3) = 4.6$, $p = .121$), suggesting more accurate internal appearance representations for the tracking models compared to humans.

Experiment 1 showed a set-size effect for humans but not for models. However, it was unclear, whether the set-size effect in humans was caused by the number of target objects or by the overall number of objects (i.e., targets and distractors). We therefore varied the *number of distractor objects* (4 or 10) independently from the number of target objects (2 or 4). First, we replicated all key findings from experiment 1 (Figure 3a&b). Figure 3c shows that human tracking performance was both a function of the number of target ($F(1, 9) = 45.4$, $p < .001$) and distractor ($F(1, 9) = 66.7$, $p < .001$) objects. Model performance in contrast was neither influenced by the number of target ($F(1, 3) = 3.0$, $p = .180$) nor the number of distractor objects ($F(1, 3) = 0.1$, $p = .822$).

# 4   Discussion and Conclusion

We compared state of the art MOT models and humans in their ability to track multiple objects through occlusion. We find that all models behaved in qualitatively different ways from human observers.

---

[1]Stimuli, data, and analysis code are available at https://github.com/Benjamin-Peters/mot-model-vs-human and https://osf.io/n7mah

[2]While showing some differences in their performance, object tracking models perform comparably similar. This is why we refrained from interpreting small differences between models. For the purpose of statistical comparison, we therefore treat the models as a sample from the distribution of supervised trackers trained on ML benchmarks.

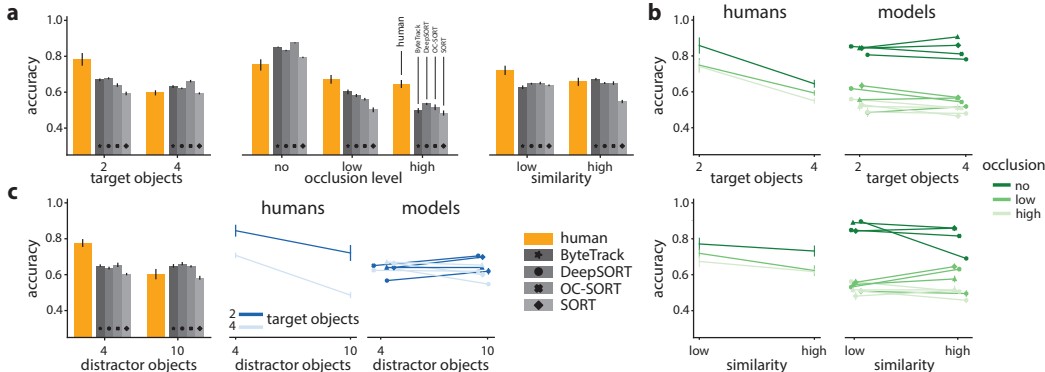

Figure 3: Results of Experiment 2. Main effects (a) and interactions (b) as in Experiment 1 (Figure 2). Main effect and interactions for the number of distractor objects (c). See supplement for more interactions and model-specific plots (Figure 6).

Most notably, without occlusion, models do not display the same capacity limitations as humans, showing superhuman performance for large numbers of targets. However, models show subhuman performance when faced with longer stretches of occlusion. These findings raise several interesting questions and future research directions suggesting combinations of normative and approximate approaches.

The normative ideal to multiple object tracking is infeasible in face of real-world complexity. Humans and models need to rely on approximate internal models of dynamics and appearance. Using systematic manipulations of object dynamics and appearance, we can reveal these approximations. For example, ML trackers were particularly challenged with the non-linear object motion under occlusion for which the linear motion model of the Kalman filter was inappropriate[3]. Future studies should vary appearance and motion models in a more principled way to reveal the approximate internal models employed by humans and ML tracking models.

The present findings are a starting point for a cognitive computational understanding of human multiple object tracking in a task space, that is situated between abstracted classical cognitive tasks and the complexity of real-world vision (Peters & Kriegeskorte, 2021). The tracking models, which we considered here, combine computationally powerful components with a conceptually low-dimensional algorithm class, in which 'what' (appearance embedding) and 'where' (bounding boxes) information is combined with internal beliefs. These models are therefore examples of "hybrid" models (Ma & Peters, 2020), that enable us to relate cognitive concepts (e.g., working memory, visual attention, or mental simulation) and phenomenology (Pylyshyn & Storm, 1988; Cavanagh & Alvarez, 2005; Oksama & Hyönä, 2004) to real-world vision and engineering.

The experimental framework introduced here promises to connect the engineering and the cognitive science of MOT and to add elements missing from both literatures: namely longer occlusions and complex motion trajectories as well as a continuum of controlled and naturalistic task variants that probe how tracking and recognition cooperate to give us a persistent representation of the objects that matter to us.

## Acknowledgments and Disclosure of Funding

B.P. has received funding from the EU Horizon 2020 research and innovation programme under the Marie Skłodowska-Curie grant agreement no. 841578. This work was also supported by the National Science Foundation under Grant No. 1948004 to N.K.

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

Figure 4: **Instruction screen**

## A   Details on the models

We evaluated tracking performance for SORT (Bewley et al., 2016), DeepSORT (Wojke et al., 2017), ByteTrack (Zhang et al., 2022), and OC-SORT (Cao et al., 2022) using the implementation and pre-trained weights from the MMTracking toolbox (Contributors, 2020). We evaluated tracking performance for all models using videos generated by the Unity framework (30 fps) (Peters et al., 2022).

## B   Data crowdsourcing

### B.1   Recruitment, procedure, and data

Participants were recruited via Prolific (ten new participants for each experiment) and the experiment was delivered via a browser app implemented in Unity (Peters et al., 2022). The study was advertised as a brief online study on Prolific (duration: 30 minutes). Upon acceptance of the experiment, participants gave informed consent (screenshot will be included in the non-anonymized version of the manuscript) and completed a brief demographic questionnaire in accordance following the procedure approved by the appropriate local IRB. The experiment then started by presenting an instruction screen (see Figure 4). After completing the experiment, participants were given a completion code with which they could verify experiment completion with Prolific and obtain their compensation. The collected data and responses on the task are non-identifiable in accordance with the IRB.

### B.2   Human participant compensation

The estimated hourly wage paid to the participants was $12. The present data amounts to 13 participant hours, i.e. a total $156 for participant compensation.

# C  Detailed view on the results

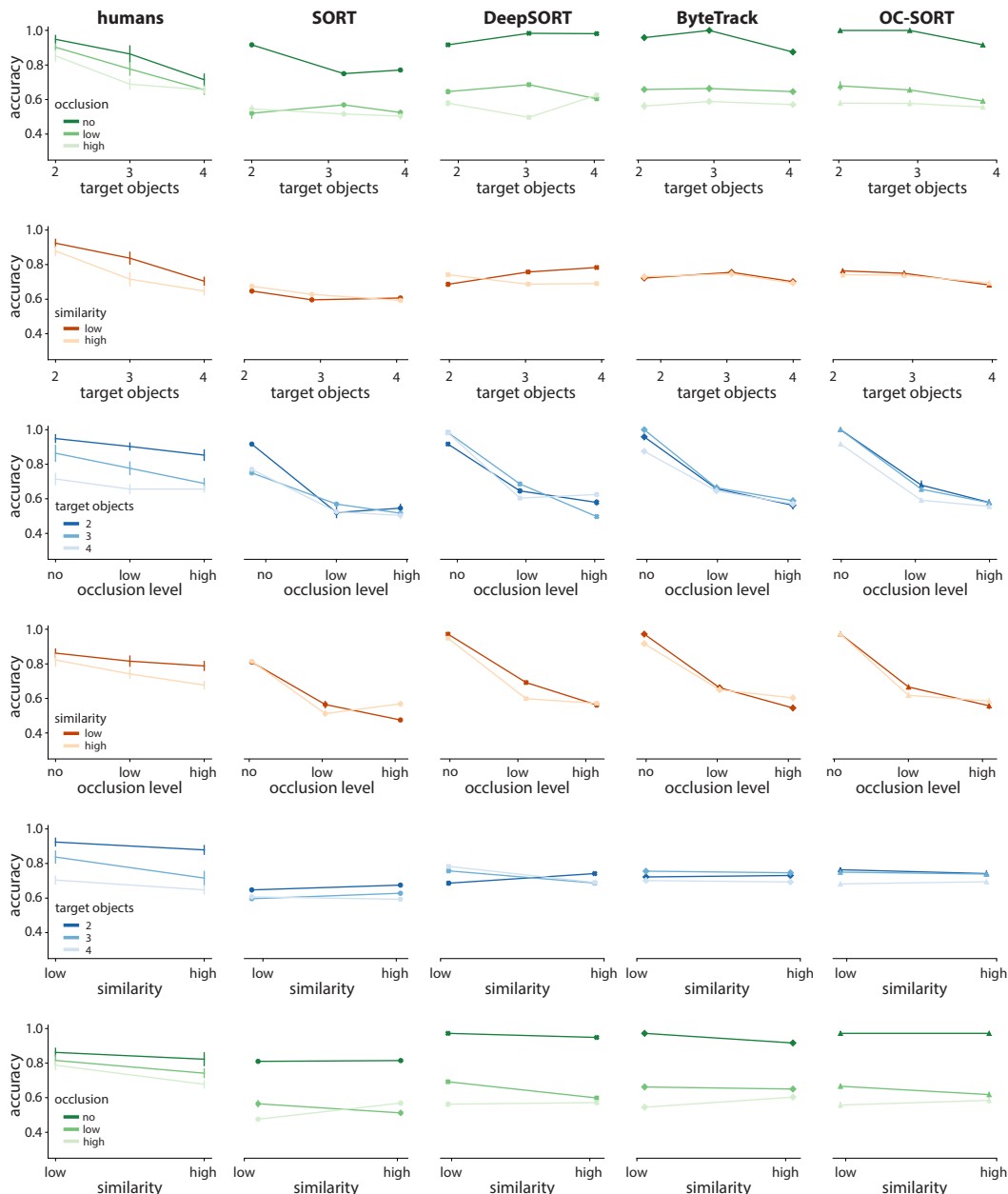

Figure 5: More detailed view on the results of Experiment 1 (Figure 2).

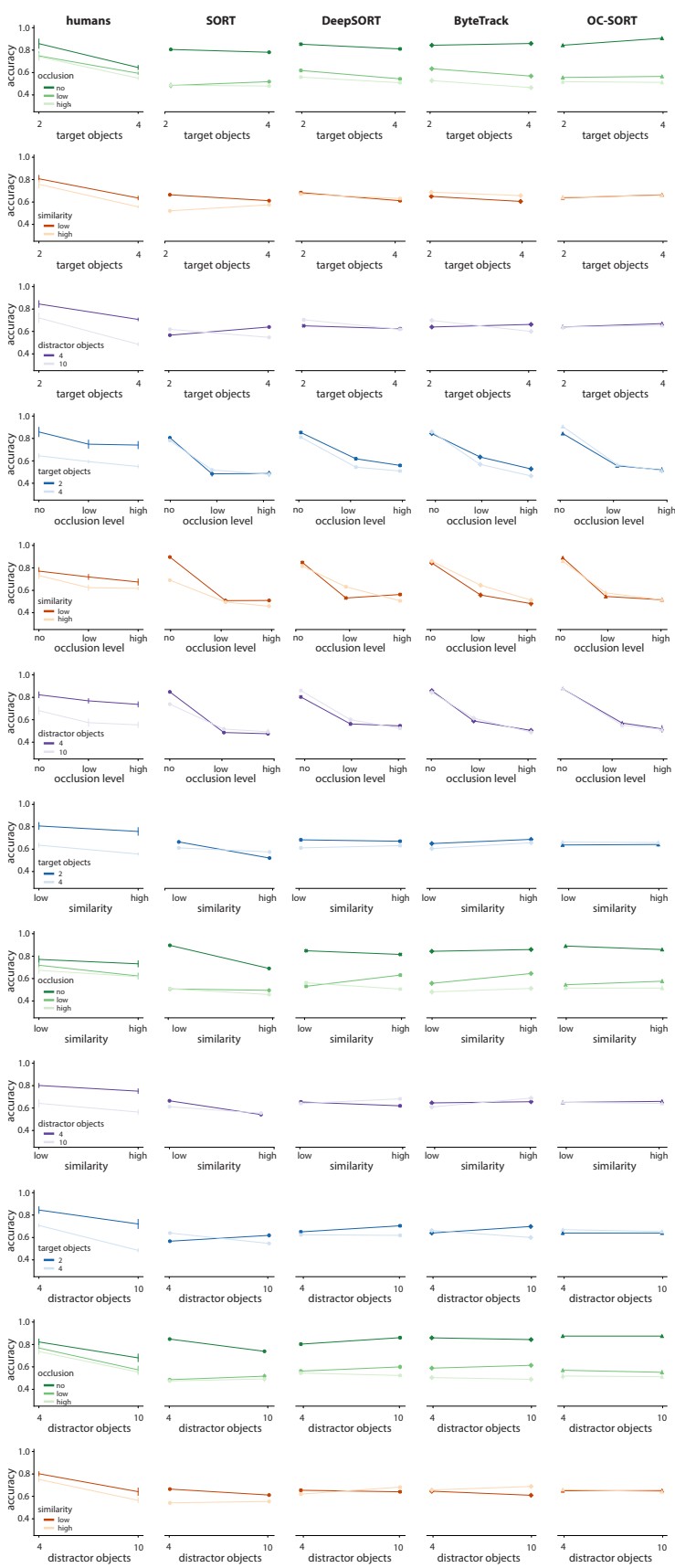

Figure 6: More detailed view on the results of Experiment 2 (Figure 3).

# D Additional model analysis

We suspected that low ML model tracking performance during occlusion was due to the linear motion model of the Kalman filter, which is inaccurate when predicting position and velocity of an occluded object with non-linear dynamics. To support this intuition, we took a closer look at DeepSORT. DeepSORT incorporates spatiotemporal predictions of objects in form of a spatial gating mechanisms: only detections, that are close enough to the position predicted by the Kalman filter are considered for the association step. The gating mechanism might therefore incorrectly exclude a detection, that emerges after a period of occlusion from behind the occluder, from the association step with the correct object. We therefore removed the spatial gating from DeepSORT and reanalyzed model behavior using the data of Experiment 1. This yielded tracking performance that was more similar to humans (Figure 7). Note however, that the spatial gating mechanism in DeepSORT serves a functional purpose (mainly reducing the computational complexity of computing distances between all detections and all tracked objects). DeepSORT 'out-of-the-box' with the spatial gating mechanism will therefore perform better and more efficiently in crowded scenarios with many brief occlusions for which the linearity assumption is "good enough" (i.e., the situation of many MOT benchmark videos, Milan et al. (2016)). These additional results suggest that a tracker should weigh internal models during the association step after occlusion according to the object-specific adequacy of these models before occlusion.

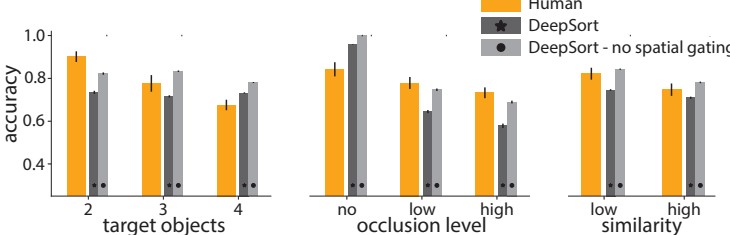

Figure 7: Removing the spatial gating from DeepSORT yields tracking performance that is closer to human tracking performance under occlusion.

