# OpenReview forum: "How do humans and machine learning models track multiple objects through occlusion?"
_NeurIPS.cc/2022/Workshop/SVRHM — SVRHM Poster_

### Official Review · Reviewer_6Xjh · 2022-10-14
**Comparing humans and CV models on MOT under occlusions.**

**Rating:** 7
**Confidence:** 3

**Review:**

quality
- have you considered comparing to models such as in ‘AGENT: A Benchmark for Core Psychological Reasoning’?
- is there a formal way in which you define object similarity?
- some implementation details are missing — number of model runs, how were objects selected from COCO in each trial

clarity
- section 2.3 could appear earlier (e.g. before 2.2.1)
- Fig2 y-axis doesn’t appear on bottom figure
- Fig3 a and b aren’t marked
- typos: line 170 ‘trough’, 349 participant*s*
- reference [6] is cut

originality
- human results are a replication of previous studies. the comparison of the human and model results under the same conditions appears to be the main contribution.

significance
- understanding CV failure cases on which humans succeed can help build human-level AI models
- results could perhaps have been somewhat anticipated (the human results replicate known results, and occlusion is a hard problem for CV models). would be interesting to gain insight from this study into why CV models fail in these cases and what seems like a promising direction for improving upon them. or, more insight into what the gaps are for natural images that have context as well.

pros
- framework for comparing humans and CV models under the same conditions

cons
- significance of insights into the gap between CV models and human performance

---

### Official Review · Reviewer_AGFQ · 2022-10-15
**Review of "How do humans and machine learning models track multiple objects through occlusion?"**

**Rating:** 7
**Confidence:** 4

**Review:**

This paper measures mutliple object tracking (MOT) in humans and machines, and compares their performance. The authors varied the tracking task along a number of parameters (number objects, number distractors, presence of occluders, target-distractor similarity). Their results replicated key human findings. For MOT computer vision models, they found that machines outperformed humans for large number of targets, underperformed for conditions with occlusions. Overall, the experiments seem well-conducted, but the project may not be novel enough for this venue.

Pros
- I found the writing to be very clear: the introduction was well-written, the summaries and descriptions of the models were clear and concise, and the description of the experiments were very clear and easy to follow.
- The authors chose a very well-motivated set of studies for establishing the benchmark of human performance in MOT. The design is simple yet they cover broad cases, allowing for a comparison between humans and models on a good overview of conditions
-The authors make a pretty good case for the need to create a benchmark of human MOT performance against which to compare computer vision models. This setup revealed some weaknesses of MOT models, in particular dealing with occlusion (although I don’t think this is a novel finding).

Cons
Overall, while this is clearly good work, it is not necessarily moving the field forward. It is a replication of human results and a reimplementation of MOT models that have already been established. The novelty in this paper is in the direct comparison to humans. These results are very interesting, and should be published, but may not be novel enough for this venue.

---

### Official Review · Reviewer_9JwP · 2022-10-16
**Tackles interesting problems in MOT, but need more interpretation of the models and results**

**Rating:** 6
**Confidence:** 3

**Review:**

The authors compare the performance of a few off-the-shelf tracking algorithms, as well as human behavior, on a multiple object tracking MOT tasks. Specifically, they test the effect of number of target object and distractors, similarity of target and distractors, and occlusion. The human behavior shows well-known effects of these parameters: performance decreases with number of target objects, similarity and occlusion. Performance was relatively unaffected by the number of distractors. Changes in model performance with the conditions seem significantly different from those of humans, both in direction and magnitude of the effect. This raises the question how similar the computations that the models perform are indeed the same as those of humans. As the authors note in the title, how (e.g. the algorithms by which) the humans and machines track is an important question, but is relatively unaddressed throughout the paper. The paper can benefit from interpreting the parameters of the models that are relevant to the task manipulations.

- Occlusion: How is the Kalman filter updated during occlusion or when there is no recognition/detection of the object? (e.g. SORT assignment rejection threshold or distance metrics etc...) This seem important to the significant decrease of performance with occlusion level. The uncertainty tracked by Kalman filter when there are no measurements increases with the dynamics and process noise assumed by the Kalman filter. What was the dynamics model used, and how much dynamics prediction error were there throughout the period of occlusion, given the nonlinear dynamics of the object?

- Why are there not much performance decrease in the models (except DeepSort) with object similarity. Does the association stage associate all detected objects regardless of their category? For instance, the algorithm may associate the banana with a bus as long as they have significant overlap in IOU? And the deepsort, since it takes into account the appearance, there may be fewer identity switches with low similarity?

- fig. 2a There is an improvement in model behavior (or no changes) in low vs high similarity. Is this difference significant, and why do you supposed this is due to?
This indicates the tracking mistakes may not be from tracklet switches (except in deepsort).
- fig.2a On a similar note, the differences may be due to averaging over confounding parameters. Are the figures averaging over the other conditions? for instance, in the first bar plot comparing target objects, the bars indicate average over all the occlusion levels and similarity levels? We might want to see the interaction between all each conditions.
- fig.2a bottom row: missing y axis.
- fig.2b the differences between the different models are hard to see
- L176-177: increased similarity between objects does not seem to reduce tracking performance in models (p = 0.5166), except perhaps in deepsort, which takes the object appearance into account
- L180: There does seem to be a small difference between the models with the number of distractors (p=0.015)

---

### Official Review · Reviewer_wgh5 · 2022-10-17
**Good generalization of previous results from toy stimuli to real-world stimuli. Very relevant to the workshop as well.**

**Rating:** 8
**Confidence:** 5

**Review:**

First off, this paper is very relevant to the workshop, has found an important gap in the existing literature, and has done a good job of filling this gap. Good job explaining the motivation behind the study- it is true that a lot of experiments were done on toy stimuli, rarely seen outside the lab, and modern techniques provide us with a way to make models that work on real-world images and not just blobs.

Here are some issues I find in the submission and some suggestions that could improve this work further:

Minor Changes:
1) Please fix the link in reference 6.
2) Figure 1c needs to be bigger.
3) The various sub-parts of Figure 3 are not labeled as 3a,3b, etc but referred to by those names in the text.

Questions/Clarifications:
1) At the response period, are the humans choosing from masked images or the actual images? Because if they can choose from actual images then this could be a memory task, not exactly a tracking task (that is, they might be using their initial memory of the target objects, instead of recent memory of the trajectories). The ability to accurately recall initially seen objects should be harder when the distractors look more similar to the target and when there are more objects (thus memory alone could potentially explain the results). So a lazy subject can just choose to ignore all the trajectory information and just remember the initial templates. To actually probe if they are indeed tracking, at the response screen, all the objects should be replaced by masks and the subjects have to click on the masks. This would require them to remember which object was going where. The text doesn’t make it super clear if this is done or not.

“During the response period, human participants clicked on T objects, which they believed to be the initially cued target objects, after which the identity of the selected objects was revealed by an animation that displayed and "collected" the coins of the target objects”- Are these T objects actual objects or icons at the location of all the objects?  "… after which the identity was revealed …" suggests that the objects were indeed masked, but Fig 1a suggests otherwise (as do lines 109 and 110).

2) Alternatively, if this is not done, at least there should be a comparison experiment where there is no motion continuity. At every time step, the locations of all the objects are changed to a random new position (so motion extrapolation gives you no cue at all and you need to rely on your memory of the targets to do the task.) If the performance varies similarly on changing the number of objects and their similarity, occlusion etc, then this would signal that humans are not really tracking and relying only on their memory.

However, to be fair, if a subject were just trying to recall the initially shown targets with no regard to motion history, I wouldn't expect them to be influenced by occlusion

3) How exactly does occlusion interfere with the model’s performance? What happens to the model parameters when it goes behind the occlusion and then re-emerges? Maybe it relies on only the last few timesteps for extrapolating trajectories while humans can use a longer past than just the last few timesteps. Is it possible to force the model to weigh long-range history more?

4) Target-distractor similarity: Belonging to the same category shouldn't be the only metric for object similarity and a more nuanced metric should be used.


Suggestions:
1) It would be interesting to see what happens if the objects are similar shape-wise but differ texture-wise? NNs could be able to pick up the texture difference while humans might not (e.g. the literature about NNs showing texture bias like Geirhos et al, 2019.)

2) This paper would be improved, in my opinion, if it integrates the visual search literature better. The set-size effect (for searching an object in a static scene) exists in the visual search literature, as well as the difficulty of finding an object when present with similar-looking distractor.

3) It would be good to tease apart the effects of three processes that could be contributing: 1) working memory load (how much of these effects would exist if motion continuity didn't exist and one had to rely only on their memory of the initial targets they were shown), 2) attention effects (for example, visual search, an attention paradigm, already shows some of these effects in static scenes, and common explanations of this set-size effect in the attention literature are resource pressure or inherent difficulty that even an ideal observer without resource pressure would face), 3) trajectory continuity.

4) The most informative points in the stimuli would be collisions between objects, and just when the occlusion boundary is passed. Maybe look into the uncertainty of the model. When the objects are clearly apart, the features of the objects would be the more useful information, but when things get harder, the continuity of motion becomes the more important information. This tradeoff would be interesting to study if you have access to the model parameters.

5) Would be interesting to implement change in orientations when the objects move. In such a case, it is possible that DNNs will show no difference if tracking an object or a face, but humans might (inverted faces are harder to tell apart for humans than objects). Further, DNNs might be worse at generalizing to new orientations but this drop in performance will not differ between objects and faces.


I have made several comments but I feel that this paper is very well-suited for this workshop and I further hope that the expanded version of this work makes it into a top journal/conference and that my reviews will help the authors improve it.